# Intracranial Epidermoid Cyst Mimics Musculoskeletal Shoulder Disease: Findings from a Case Report in Physiotherapy Clinical Practice

**DOI:** 10.3390/ijerph192013590

**Published:** 2022-10-20

**Authors:** Fabrizio Brindisino, Mariangela Lorusso, Lorenza De Carlo, Firas Mourad, Sharon Marruganti, Valerio Passudetti, Mattia Salomon

**Affiliations:** 1Department of Medicine and Health Science “Vincenzo Tiberio”, University of Molise C/da Tappino c/o Cardarelli Hospital, 86100 Campobasso, Italy; 2Department of Clinical Sciences and Translational Medicine, University of Roma “Tor Vergata” c/o Medicine and Surgery School, 00133 Rome, Italy; 3Fisioworld Private Practice, 86021 Bojano, Italy; 4Department of Physiotherapy, LUNEX International University of Health, Exercise and Sports, 4671 Luxembourg, Luxembourg; 5Luxembourg Health & Sport Sciences Research Institute A.s.b.l., 50, Avenue du Parc des Sports, 4671 Luxembourg, Luxembourg

**Keywords:** differential diagnosis, epidermoid cyst, physical therapy modalities, shoulder pain

## Abstract

Shoulder pain is often attributable to a musculoskeletal disorder, but in some instances, it may be linked to pathologies outside the physiotherapist’s area of expertise. Specifically, some intracranial problems can cause pain and disability to the shoulder complex. This case report aims to describe the clinical presentation, history taking, physical examination, and clinical decision-making procedures in a patient with an intracranial epidermoid cyst mimicking a musculoskeletal disorder of the shoulder girdle. A 42-year-old man complained of pain and disability in his left shoulder. Sudden, sharp pain was reported during overhead movements, associated with intermittent tingling of the left upper trapezius and left scapular area. Moreover, the patient reported reduced hearing in his left ear and left facial dysesthesia. The physical examination led the physiotherapist to hypothesize a pathology outside the physiotherapist’s scope of practice and to refer the patient to another health professional to further investigate the patient through imaging. It is essential for the physiotherapist to recognize when the patient’s clinical condition requires a referral to another healthcare professional. Therefore, the physiotherapist must be able to, in a timely manner, identify signs and symptoms suggesting the presence of medical pathology beyond his expertise, through appropriate medical history collection and physical evaluation.

## 1. Introduction

The most important ability for a physiotherapist (PT), especially when working in direct access, is to make sure that each patient is an appropriate candidate for physiotherapy treatment [1,2].

To do so, the PT needs to be able to identify signs and symptoms of pathologies that are potentially outside the PT’s scope of practice [1,2]. In fact, the differential screening process is counted among the PT’s essential skills [3,4], aiming to identify signs and symptoms of pathologies outside the rehabilitation interest (e.g., tumours, infections) that require prompt referral and adequate management [5].

Notably, shoulder pain can be caused by many non-neuro-musculoskeletal disorders [3,6,7,8,9]. In this context, some rare intracranial pathologies, in particular, can affect the shoulder complex in terms of pain and disability [6,10]; they are often characterized by compressive space-occupying masses [7] and/or ischemia in areas that are functionally connected to the shoulder [10]. Specifically, most of these compression phenomena may affect structures that contribute to the innervation of the shoulder muscles, causing motor deficits and decreased functional capacity.

First described by Cruveilhier in 1835 [11], the epidermoid cyst may manifest at any age, but it is more frequent in male adults [12,13,14]. An epidermoid cyst could be located in any part of the body, although most commonly in the face, neck, chest, upper back, and genital areas. Rarely present in the central nervous system or in the intracranial region [12], with an incidence of between 0.3% and 1.8% of all brain tumours [15], an epidermoid cyst is recognized as a benign tumour, becoming malignant in 1% of cases [16,17].

The prevailing localization of intracranial epidermoid cyst is the cerebellopontine angle (37.3%), followed by the parasellar region (30%) and the middle cranial fossa (18%) [15]. Usually, these tumours grow insidiously, with a linear speed, causing a very slow symptomatic onset [15] and only occasionally mass effect, cranial neuropathies, convulsions [18], and/or other manifestations based on the location. Lastly, rupture of the epidermoid cyst can cause aseptic granulomatous meningitis [19]. Correlation between the sites of lesion and potential symptoms referred by previous studies is reported in Table 1.

Diagnosis of intracranial epidermoid cyst is made via medical imaging, such as computed tomography (CT) or magnetic resonance imaging (MRI). Patients with asymptomatic epidermoid cyst need regular medical evaluation with imaging tests. However, for symptomatic patients, the best choice [19] is surgery, because it is essential to perform radical resection [18] to prevent recurrence [15].

As far as the authors know, no clinical case has yet reported a clinical presentation and physical examination that describes a patient with intracranial epidermoid cyst mimicking a neuro-musculoskeletal disorder of the shoulder complex assessed in physiotherapy direct access clinical practice. For this reason, this case report describes the physiotherapist’s clinical decision-making process, documenting the importance of screening for referral for a better and prompt clinical management of the patient.

## 2. Case Presentation

This case report was written following the CARE guidelines [22]. The patient described in this clinical case authorized the authors to report his case through written informed consent. The clinical history and key features are summarized in the timeline (Figure 1).

The patient, a 42-year-old right-handed entrepreneur and amateur cyclist for about 10 years (training four times a week for about 50 km per session), went to the author’s private practice complaining of pain and disability in his left shoulder.

The patient reported a constant and nagging pain, localized on the left side of the neck, down to midarm (5/10 Numeric Pain Rating Scale (NPRS)); furthermore, the patient reported pain at the scapula (6/10 NPRS), under the axilla (2/10 NPRS) (Figure 2—body chart), and a significant sharp pain in the shoulder (8/10 NPRS) such as to make it impossible for him to raise his arm over his head.

The patient reported suffering from chronic neck pain; however, in the last month, the neck pain was reported to have changed, becoming persistent and associated with intermittent paresthesia on the upper trapezius and, on the back, on the scapular area, warned as a pinprick sensation both during everyday activities (e.g., lifting c. 5 kg objects) and while resting in bed. The patient declared that the pain arose in the neck at first, about a month before the consultation, with low intensity (2/10 NPRS). However, the pain gradually increased (5/10 NPRS) and diffused, wrapping the upper trapezius and the entire shoulder complex. The pain increased over time and, progressively, the arm function decreased; originally, the patient felt that the arm had only weakened, but later he was no longer able to raise the arm over his head.

The patient declared having never experienced similar episodes of pain before; moreover, the drugs taken to relieve pain (ibuprofen and tramadol) were not effective in reducing the symptoms. The patient reported that, at 20 days after the onset of symptoms, he had been forced to interrupt his activity as a cyclist because of arm pain and weakness; moreover, the pain also affected the activities of daily life and work.

The patient stated that, because of pain and associated disorders, his sleep was no longer restful; in fact, the pain often woke him up, and no position could relieve it. Furthermore, the patient was concerned because the pain, from being low in intensity and dull, had become worse, was cramping and deep (Figure 2—body chart), and had a negative influence on his working and sporting life.

A deeper clinical investigation regarding associated symptoms, as suggested by the evidence [8,23], led the PT to question the patient about any other significant disturbance or discomfort, but no other neuro-musculoskeletal disorders were reported by the patient. A checklist for significant or potential “red flags” as a widely and accepted system to rule out serious pathologies or systemic symptoms (fever, weight loss, balance problems) was performed [3,24,25,26]. The patient did not report any previous injury or surgical interventions, other pathologies, or other relevant genetic information, except for mild intermittent paresthesia of the left cheek and jaw and that he noticed a progressive reduction in hearing in his left ear.

## 3. Clinical Examination

The PT chose first to assess the symptoms complained about by the patient (shoulder pain and functional deficit), and then, so as to further investigate issues arising from the medical history, to continue the evaluation assessing jaw paresthesia and hearing loss.

### 3.1. Neuro-Musculoskeletal Assessment

The PT began the clinical examination observing the patient on the frontal (anterior and posterior) and sagittal planes, looking for asymmetries, postural attitude of the patient, and muscular trophism. The observation on the anterior frontal plane revealed a depressed left shoulder, the left clavicle more protruding, and the left upper trapezius and left side of the neck more atrophic and hypotonic than the right. The posterior frontal observation revealed an “empty” left supraspinous space, internally rotated left arm, and an abducted and downwards rotated left scapula; moreover, the space between the thoracic spine and the medial edge of the left scapula looked hypotonic compared with the contralateral. The muscle trophism of the back muscles was symmetrical, except for the muscles between the thoracic spine and the medial scapular edge and the left upper trapezius, which appeared severely atrophic and hypotonic.

After this, the PT investigated whether these postural alterations of the shoulder girdle and muscle trophism influenced on the patient’s active and passive movement. A shoulder range of motion (ROM) assessment was performed using a Tracker Freedom wireless inclinometer (JTECH Medical, Midavele, UT, USA) [27]. The patient was asked to move the arm three times in a particular plane and an average value was reported. Active right ROM was complete in flexion and abduction, while active left shoulder flexion was 75° and abduction was 80°. Performing these movements, left scapulo-thoracic rhythm was altered: the left scapula rotated more internally and downward than the right; in addition, the left scapular medial edge was protruding. Owing to severe pain, the patient could not perform left overhead movements (abduction and flexion 8/10 NPRS). Left shoulder passive ROM was the same as the contralateral in all movements; only the last parts of left shoulder flexion and abduction were mildly painful (2/10 NPRS).

Manual shoulder assessment with shoulder strength testing was performed, with isometric resistance using the standard position and procedures described in the literature [28], aiming to screen the isolated muscle condition and ability through the Kendall scale [29]. The Kendall scale is a method that tests a specific muscle action orienting the body parts in a particular direction to selectively assess a particular muscle’s load capacity. There is evidence for good reliability and validity in the use of manual muscle tests for patients with neuromusculoskeletal dysfunction [30], and the Kendall scale was also chosen because it may also provide results regarding a particular muscle’s weakness and the awareness of possible substitution by other, stronger muscles [29].

The PT measured left upper trapezius muscle strength by asking the seated patient to raise both shoulders [28]. Then, the PT measured the left middle/lower trapezius muscle strength by asking the patient, lying on his right side, to perform a horizontal abduction with the left shoulder flexed at 90° and elbow extended [28]. Scoring was calculated using the Kendall scale [29]: the PT attributed a 1/5 score both to left upper and middle/lower trapezius, while right upper and middle/lower trapezius muscle strength was scored as 5/5 [29].

The PT measured left sternocleidomastoid muscle strength by asking the patient to stay in the test position (the patient lay on his right side with head in flexion, left lateral flexion, and contralateral rotation [28]); the PT scored the left sternocleidomastoid muscle as 2/5, while the right was scored as 5/5 using the Kendal scale [29].

Other muscles were tested (anterior, posterior and middle deltoid, serratus anterior, biceps and triceps brachii, pectoralis major, latissimus dorsi, rotator cuff muscles, and the muscle of the temporomandibular joint), but no differences in strength regarding the right side were found.

The muscle testing found a clear isolated weakness of the left trapezius and sternocleidomastoid muscles, suggesting that the patient’s complaint significantly impacted observed muscle trophism, posture, and active ROM, as well as the strength of a part of the left shoulder girdle.

Owing to the previous history of paresthesia on the upper trapezius and lateral scapular area, the PT also investigated neck function [31]. On both sides, a cervical lateral glide test as well as a compression and distraction test were performed, but no clinical features were revealed. Moreover, a quick neurological examination and quantitative sensory testing [25] were administered with no significant results.

### 3.2. Cranial Nerve Assessment

In order to better understand the patient’s unclear cranial and facial symptoms and left-ear hearing loss, the PT also performed a cranial nerve assessment [32]. This examination was easily and quickly performed with a Snellen chart, together with a pen light or small flashlight, neurotips, cotton wool, and tongue depressor [33] (Appendix A). All of the examinations apart from V and VIII cranic nerves had negative results.

### 3.3. V Nerve—Trigeminal (Small Sensory Nerve Fibers’ Evaluation)

The PT evaluated patient-reported facial dysesthesia by assessing the presence of any disturbances in electrical conduction of small-caliber sensory nerve fibers. To be specific, the PT tested the patient’s pain response using pinprick stimulation and thermal sensitivity [34].

A neurotip (Owen-Munford Neuropen) was used to evaluate the patient’s pain response to a standardized noxious stimulus [34]; first, the PT placed the neurotip in the centre of the body of the left jaw, increasing the pressure on the skin, without piercing it, until the skin turned pale. The same assessment was performed for the right jaw. To the PT’s question as to whether these two painful stimuli were comparable, the patient underlined a reduced sensitivity on the left, suggesting a reduced mechanical pain threshold on that side of the face [34]. The next step was to evaluate the ability to discriminate thermal sensations using two coins.

A metal coin is a good heat conductor and it is perceived as ‘cold’ at room temperature in a healthy population, whereas a metal coin placed in the pocket is perceived as neutral or slightly warm in a healthy population [34].

To assess the patient’s cold detection, the PT used a coin kept at room temperature, placing this first on the right cheek and then on the left one. To assess patient heat detection ability, the PT used a coin kept in his pocket for about 30 min, repeating the same procedure described for the cold one.

When asked by the PT if any thermal difference was detected between the two sides, the patient replied to have perceived the room temperature coin as less cold when placed on the left side than when placed on the right one, suggesting a deficit in cold detection. On the other hand, with the coin previously kept in a pocket, the patient replied to have perceived it as colder on the left side than on the right, suggesting an impairment of the ability to detect heat stimuli [34].

### 3.4. VIII Vestibulocochlear/Auditory (Hearing Screening Test)

In order to rule out hearing impairment, the PT performed the “finger rub test” and the “whispered voice test” [35], which are the most accurate and easy-to-use assessment tests in an outpatient setting among the tools for hearing screening [35].

The “finger rub test” is performed by gently rubbing six times the fingers about 15 cm from the patient’s ear, while the patient is seated with the contralateral ear plugged. The test is positive when the patient is unable to identify the sound of rubbing fingers at least three out of the six times [35].

The “whispered voice test” is performed by whispering six times different combinations of numbers, being placed about 60 cm behind the patient, while he is seated with the opposite ear to be tested plugged. The test is positive when the patient is unable to repeat whispered numbers at least three out of the six times [35].

The patient in this case report could not identify finger rubbing sounds five out of the six times and could not repeat the whispered numbers correctly four out of the six times when left ear functionality was tested, suggesting a hearing loss of approximately 25–30 decibels [35].

The assessed muscle strength and trophism loss in a short time and without any known cause, in addition to small sensory nerve fibers’ evaluation and hearing screening tests results, suggested that the patient’s symptoms did not have a neuro-musculoskeletal origin. The PT hypothesized a compression of the brainstem, probably localized in the nuclei of the accessory (XI°), facial (VII°), and vestibulocochlear (VIII°) nerves.

### 3.5. Patient-Reported Outcome Measures—Assessment at Baseline

To specifically evaluate the patient’s health status, the PT administered the Italian version of the Shoulder Pain and Disability Index (SPADI) [36] as a specific assessment tool for the shoulder, and the Italian version of the Short Form-36 Health Questionnaire (SF-36) in order to assess the patient’s quality of life [37]. The results are summarized in Table 2.

Applying the International Classification of Functioning, Disability, and Health (ICF) framework to the SF-36 evaluation revealed that the most compromised domains were “participation” and “perceived health and functioning” [38].

### 3.6. Referral

After administering the questionnaires, the PT referred the patient to a neurosurgeon consultant with a detailed letter on the main data collected through the medical history and clinical examination.

The neurosurgeon prescribed an MRI with contrast medium (gadolinium) and an electromyography. The MRI revealed an “area of altered signal (diameter of 5.2 × 2.6 × 3 cm) at the level of the left cerebellopontine angle, which had a coarse fibrotic sprout and micro sprouts inside, with a chaotic distribution. The intralesional fluid content was not very homogenous and had fine corpuscles. This cystic-type formation significantly compressed the brain stem, which was displaced to the right of the midline, and pushed the angle of the ipsilateral cerebellar peduncle in a postero-medial direction, deforming it. Furthermore, the formation compressed the left-anterior part of the pons, close to the basilar artery” (Figure 3). The electromyography/elettroneurography examination showed severe axonal damage along the motor fibers of the left accessory (XI) and suprascapular nerves. In particular, the traces at maximum effort are defined as “interferential” on the deltoid muscles and biceps brachialis, while they are “poor-intermediate” on the supraspinatus and trapezius muscles. Such assessment also showed an initial slowdown in the distal motor latency and in the sensory conduction velocity of the left median nerve due to antidromic recording from the third finger of the left hand. The sensory and motor neurography of the left radial and ulnar nerve was within the limits and the neurography of the left musculocutaneous nerve was also within the limits, while the amplitude of the compound muscle action potential of the left accessory nerve was reduced compared with the contralateral. Therefore, the neurosurgeon recommended surgery.

## 4. Surgical Procedure

A craniotomy of the left temporal bone was performed, and the cranial nerves were detected and preserved, thus avoiding any damage to them. The tumor was exposed and completely excised with its capsule to prevent recurrence (Figure 4).

Macroscopically, the tumor was pearl white, and its walls were made of stratified squamous epithelium and connective tissue. Histological examination of the tumor confirmed the diagnosis of epidermoid cyst. During the postoperative course, there were no adverse events and, a week after surgery, the patient was discharged from the hospital.

Forty days after surgery, the patient started post-surgical rehabilitation, as prescribed by the surgeon (Appendix B).

### Patient-Reported Outcome Measures—Follow-Up Assessment

The PT reassessed the patient 5 weeks after the start of rehabilitation treatment: all scores were improved from baseline; in particular, SPADI values were higher than the minimal clinically important difference (Table 3). After 5 months, the patient was back working and cycling at pre-surgery levels.

## 5. Discussion

In this case report, the patient’s clinical presentation and physical examination led the PT to detect a cluster of red flags and to hypothesize that the patient’s complaints had a non-musculoskeletal origin and that additional skills were necessary to confirm the likely existence of a pathology outside of physiotherapy practice.

A neuroanatomical reason for an intracranial cyst of the posterior fossa should cause shoulder pain (alongside motor dysfunction, muscle atrophy, hear loss, and sensitive symptoms), deserving a deeper explanation. Moreover, the combination of subtle facial reported symptoms and the important loss of function of the upper limb, mimicking much more common musculoskeletal disorders, are dissimilar from what is reported in the literature. Plausibly, the cyst significantly compressed the brain stem in the posteromedial corner. The compression of this area caused the alteration of part of cranial nerve function, specifically the vestibular-cochlear nerve, with consequent onset of hearing loss; the facial nerve, causing the appearance of check and jaw dysesthesia; and the spinal accessory nerve. This latter nerve innervates the trapezius muscles that work in synergy with the serratus anterior muscle for lifting the upper limb. Thus, the cyst that compressed the spinal accessory nerve caused an alteration to shoulder biomechanics, triggering an imbalance between the trapezius and the serratus anterior muscles. This led to a compression of the structures present at the level of the coraco-acromial arch, which probably caused the onset of shoulder pain and dysfunction.

In this case report, clinical reasoning scrutinized the particular features of symptoms and their change and progression (from nagging to cramping and severe); the atraumatic and insidious onset; the paresthesia felt in the upper trapezius, cheek, and jaw; the alleviating and aggravating factors; the reduction in muscle strength and function; and the absence of restful sleep led the PT to perform special tests for a more in-depth evaluation [39]. In fact, all of these features were integrated with the results of the small fibers’ tests for face epicritic and thermal sensitivity and hearing tests [34,35], and convinced the PT to refer the patient.

The authors think that the combination of subtle facial reported symptoms and important loss of function of the upper limb, mimicking much more common musculoskeletal disorders, are dissimilar from what is usually reported in literature. The most plausible mechanism that could justify this atypical presentation is described above. Moreover, this particular combination of symptoms, to the best of the authors’ knowledge, has not yet been reported.

The evidence confirms that the prompt detection of red flags and a subsequent timely and appropriate patient referral can help to diagnose many non-musculoskeletal medical pathologies, and this clinical case emphasizes how this best practice is fundamental for finding the most suitable treatment for the patient [3], as a later diagnosis could have led to a worse prognosis. Therefore, considering that many PTs work under direct access, it is essential that these health care professionals know when the patient must be referred [3]. The patient’s clinical data, which are collected mostly through the anamnesis, influence the PT’s therapeutic action and are crucial for a correct PT functional diagnosis [4]. The interview represents the milestone of the evaluation because it allows the examiner to detect important elements, modulating and guiding the clinical approach, and influencing the patient’s prognosis. Therefore, taking a proper, detailed, and thorough anamnestic collection should be the healthcare professional’s mandatory skill [40].

The importance of direct access in physiotherapy private practice is emphasized in many scientific studies [41,42]; this allows a considerable saving in terms of time in the overall care of the patient, fewer diagnostic tests and drugs’ prescriptions, and a more appropriate referral in the case of pathology outside physiotherapy’s scope of practice. Moreover, patients screened in physiotherapy direct access are more likely to report a high level of satisfaction [41]. To solve this specific case, better and more refined screening abilities were necessary [8], and these were acquired by attending post-graduate qualification; in fact, a bachelor’s degree basic education does not seem to ensure this expertise [43].

An Italian study investigated the importance of PT expertise in cranial nerve screening and concluded that Italian PTs were not able to perform an appropriate cranial nerve screening in the clinical practice probably because this was not a focus point of teaching in the bachelor’s degree. This knowledge, then, should be a key skill included in the PT’s core curriculum [44].

## 6. Conclusions

In this case report, post-surgical rehabilitation was crucial not only from a physical standpoint, leading the patient to recover muscle strength, proprioception, and motor coordination, but also from a psycho-social standpoint, as the patient was greatly concerned by his physical condition and was afraid that he would be forced to limit his everyday life and give up his sports activities. There is no evidence related to post-surgical rehabilitation after epidermoid cyst excision; therefore, rehabilitation was set on the specific recovery rhythms of the patient, in his compliance with supervised treatment and home exercises. Furthermore, to provide the patient bio-psycho-social support, the PT considered the patient’s preferences and expectations, personal characteristics, cultural orientations, beliefs, and experiences, involving the patient in the decision-making process [45] with the aim to promote his self-efficacy and to increase participation, motivation, and therapeutic adherence.

## Figures and Tables

**Figure 1 ijerph-19-13590-f001:**
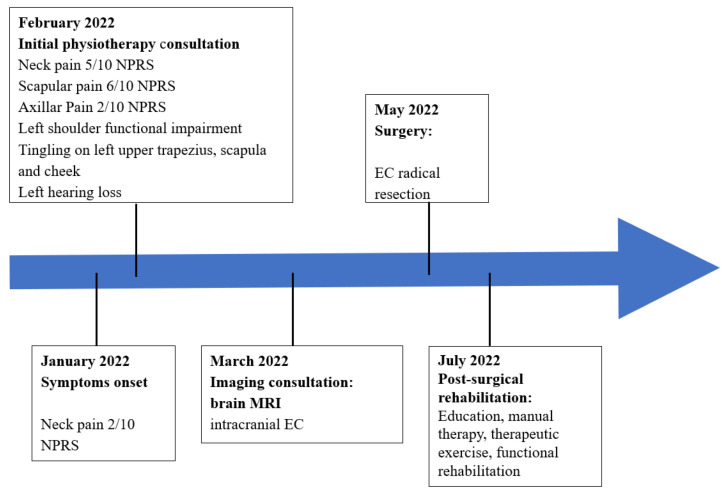
Timeline. Acronyms: MRI: magnetic resonance imaging; EC: epidermoid cyst; NPRS: Numeric Pain Rating Scale.

**Figure 2 ijerph-19-13590-f002:**
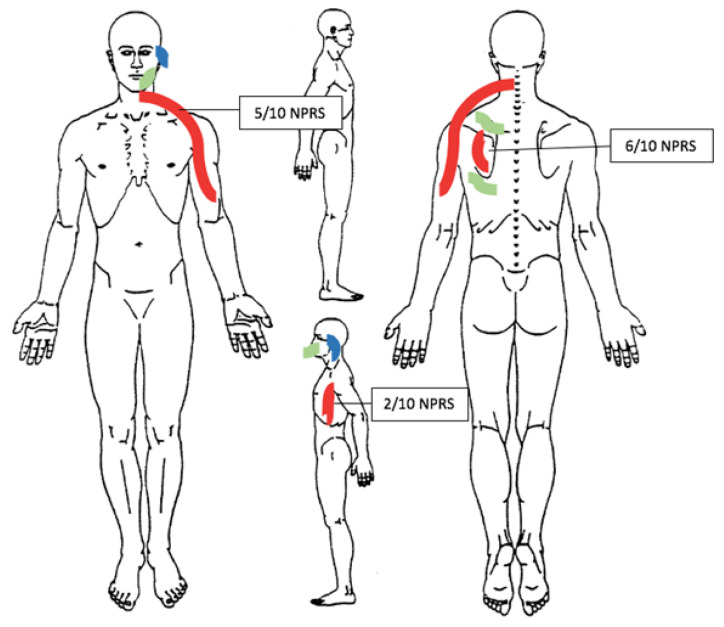
Body chart. Notes: Red lines indicate pain areas; green lines indicate paresthesia; blue lines indicate hearing impairment.

**Figure 3 ijerph-19-13590-f003:**
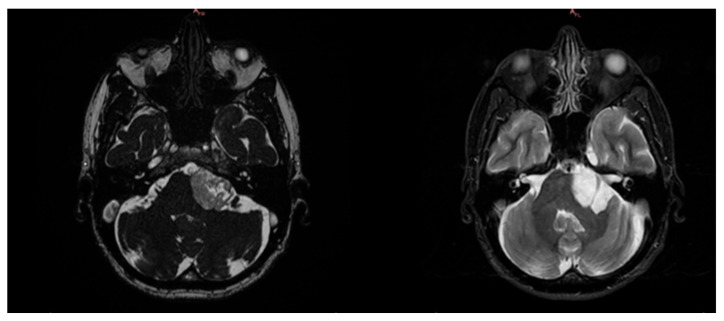
MRI scans before surgery. Notes: T1 weighted sequence (**left**) and SE-T2 weighted scan (**right**). Cystic type formation compressing the brain stem and deforming the angle of the ipsilateral cerebellar peduncle.

**Figure 4 ijerph-19-13590-f004:**
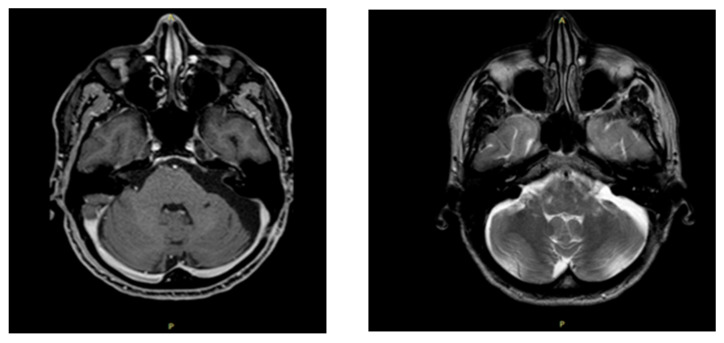
MRI scans after surgery. Notes: T1 weighted sequence (**left**) and SE-T2 weighted scan (**right**). Postoperative MRI scans show complete tumor excision.

**Table 1 ijerph-19-13590-t001:** Localization of the intracranial epidermoid cyst and related signs and symptoms.

Localization	Related Signs and Symptoms
Right temporal lobe [15]	Headache
Right parietal lobe [17]	Temporary loss of consciousness; epilepsy.
Left cerebellar cistern [20]	Dizziness; diplopia; lateral gaze palsy due to left sixth cranial nerve palsy.
Left cerebellopontine angle [12]	Dizziness; cerebellar ataxia; left facial paralysis.
Right and left frontal lobe [16]	Headache during the day that gets worse at night. Vomiting; nausea; amnesia; balance diseases; sphincter dysfunction in the form of stress incontinence.
Pineal gland [21]	Diplopia; headache; pain and stiffness in the neck; nystagmus; papilledema

**Table 2 ijerph-19-13590-t002:** Outcome measure. Baseline assessment.

**SF-36**							
**PCS**				**MCS**			
PF	RP	BP	GH	VT	RE	MH	SF
65	0	22	25	20	25	0	32
SPADI							
**Pain subscale ^a^**				**Disability subscale ^b^**			
42				70			
**NPRS**							
**Cervical Spine**			**Scapula**			**Axillar Area**	
5/10			6/10			2/10	
**ROM MEASUREMENT**							
**Active motions**							
Left shoulder flexion			Left shoulder abduction			Raising left arm above head	
75° (7/10 NPRS)			80° (7/10 NPRS)			na	
**Passive motions**							
Left shoulder flexion		Left shoulder extension		Left shoulder abduction		Left shoulder adduction	
Equal to right shoulder ROM		Equal to right shoulder ROM		Equal to right shoulder ROM		Equal to right shoulder ROM	
Painful end-range (2/10 NPRS)		-		Painful end-range (2/10 NPRS)		-	
**Small sensory nerve fibers evaluation**							
**Neurotip test**			**Heat detection**			**Cold detection**	
Reduced pain sensitivity on left jaw			Reduced heat sensitivity on left jaw			Reduced cold sensitivity on left jaw	
**Hearing testing**							
Finger rib test				Whispered voice test			
The patient could identify the sounds 1 out of 6 times				The patient could repeat the whispered numbers correctly 2 out of 6 times			

Acronyms: SF-36: Short Form Health Survey, Range 0–100 (0 = less quality of life, 100 = better quality of life); PCS: physical component summary; MCS: mental component summary; PF: physical functioning; RP: role—physical; BP: bodily pain; GH: general health; VT: vitality; SF: social functioning; RE: role—emotional; MH: mental health; ROM: range of movement; SPADI: Shoulder Pain and Disability Index, ^a^ Range 0–50 (0 = no pain; 50 = worst pain) and ^b^ Range 0–80 (0 = no disability; 80 = worst disability); NPRS: Numeric Pain Rating Scale, Range 0–10 (0 = no pain; 10 = worst pain); na: not available.

**Table 3 ijerph-19-13590-t003:** Outcome measure. Follow-up assessment.

**SF-36**							
**PCS**				**MCS**			
PF	RP	BP	GH	VT	RE	MH	SF
95	50	74	67	65	75	100	84
SPADI							
**Pain subscale ^a^**				**Disability subscale ^b^**			
12				24			
**NPRS**							
**Cervical Spine**			**Scapula**			**Axillar area**	
2/10			3/10			0/10	
**ROM Measurement**							
**Active motions**							
Left shoulder flexion			Left shoulder abduction			Raising left arm above head	
165° (1/10 NPRS)			110° (2/10 NPRS)			Performable	
**Passive motions**							
Left shoulder flexion		Left shoulder extension		Left shoulder abduction		Left shoulder adduction	
Equal to right shoulder ROM		Equal to right shoulder ROM		Equal to right shoulder ROM		Equal to right shoulder ROM	
-		-		-		-	
**Small sensory nerve fibers’ evaluation**							
**Neurotip test**			**Heat detection**			**Cold detection**	
Reduced pain sensitivity on left jaw			Equal heat sensitivity, both left/right jaws			Equal cold sensitivity, both left/right jaws	
**Hearing testing**							
Finger rib test				Whispered voice test			
The patient could identify the sounds 4 out of 6 times				The patient could repeat the whispered numbers correctly 4 out of 6 times			

Acronyms: SF-36: Short Form Health Survey, Range 0–100 (0 = less quality of life, 100 = better quality of life); PCS: physical component summary; MCS: mental component summary; PF: physical functioning; RP: role—physical; BP: bodily pain; GH: general health; VT: vitality; SF: social functioning; RE: role—emotional; MH: mental health; ROM: range of movement; SPADI: Shoulder Pain and Disability Index, ^a^ Range 0–50 (0 = no pain; 50 = worst pain) and ^b^ Range 0–80 (0 = no disability; 80 = worst disability); NPRS: Numeric Pain Rating Scale, Range 0–10 (0 = no pain; 10 = worst pain); na: not available.

## Data Availability

Not applicable.

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
