# Peer review of "Intracranial Epidermoid Cyst Mimics Musculoskeletal Shoulder Disease: Findings from a Case Report in Physiotherapy Clinical Practice"

_ijerph, 2022, doi:10.3390/ijerph192013590_

Round 1
Reviewer 1 Report
In this paper, the authors submit an interesting case of a patient with an intracranial epidermoid cyst mimicking a musculoskeletal disorder of the shoulder. This paper is well described in case presentation of medical history and physical evaluation and can confirm accurate diagnosis and treatment based on detailed physical examination and neurological examination including cranial nerve. It is also considered an important case in physiotherapy practice.
Case reports are considered for a rare condition reported, atypical symptoms and signs are observed, new diagnostic or therapeutic methods are introduced, or atypical clinical findings. This manuscript cannot confirm these.
Author Response
REVIEWER 1
We would like to thank you and the reviewer 1 for the thoughtful feedback provided on our Manuscript entitled “Intracranial epidermoid cyst mimics musculoskeletal shoulder disease: findings from a case report in physiotherapy clinical practice”. Thank you for the opportunity to revise and re-submit.
In this paper, the authors submit an interesting case of a patient with an intracranial epidermoid cyst mimicking a musculoskeletal disorder of the shoulder. This paper is well described in case presentation of medical history and physical evaluation and can confirm accurate diagnosis and treatment based on detailed physical examination and neurological examination including cranial nerve. It is also considered an important case in physiotherapy practice.
Case reports are considered for a rare condition reported, atypical symptoms and signs are observed, new diagnostic or therapeutic methods are introduced, or atypical clinical findings. This manuscript cannot confirm these.
REPLY: We would like to thank reviewer 1 for his suggestions and consideration. We really appreciate the statement about case description and physical examination procedures adopted for this particular case. The authors would like to emphasize once again the uniqueness of this case. As quoted in the text, the epidermoid cyst may manifest at any age, but rarely it is present in the central nervous system or in the intracranial region, with an incidence rate between 0.3% and 1.8% of all brain tumours.
In this epidemiological context, the rarity of this tumor acquires more importance due to the atypical clinical presentation described within the manuscript: as far as the authors know, no clinical case has yet been reported the clinical presentation and physical examination which describes a patient with intracranial epidermoid cyst mimicking a neuro-musculoskeletal disorder of the shoulder complex assessed in physiotherapy direct access clinical practice. In details, the presence of both central/cranial nerves involvement and upper limb signs and symptoms are rare. A thorough neuro-musculoskeletal and specific cranial nerve assessment led the physiotherapist to suspect the possible compression of the brainstem structures, probably localized in the nuclei of the accessory (XI°), facial (VII°) and vestibulocochlear (VIII°) nerves.
So, even if further diagnostic assessment procedures (imaging) and therapeutic methods are more common and well recognized, timely and complete screening for referral processes in presence of atypical and subtle clinical finding are the strength of this manuscript. So, as cited by Vandenbroucke in 2001, we hope that this case reports could have a high sensitivity for detecting novelty, being kind of a cornerstone of medical progress, as far as musculoskeletal disorders are concerned.
Reviewer 2 Report
This is an interesting case report with prevalent relevance for clinical practice of physical therapists, describing a case of intracranial epidermoid cyst causing shoulder pain. An accurate physical examination by the PT was able to detect red flags, causing referral to MRI and consequent surgery.
Although the take-home message is clear and nicely discussed, there are some points that need further clarification.
1- Table 1 is misleading in the way it was conceived. The authors should specify that the correlation between site of localization and symptoms is only based on previous reports, as for example a compressive lesion in the right parietal lobe could theoretically cause a wide range of symptoms other than epilepsy.
2- Figure 1: precise timing between consequent events should be specified in the timeline.
3-Paragraph 3.6 Referral: a neurological physical examination and precise quantitative EMG/ENG data are lacking from case description. In particular, electromiographic and nerve conduction findings should be described separately and with greater detail.
4- Figures 4 and 5: legends should specify which sequences are reported. Please, consider using the same sequences for pre- and post-operative scans,.
5- Discussion: a neuroanatomical explanation of why intracranial cyst of the posterior fossa should cause shoulder pain (alongside with motor dysfunction, muscle atrophy, hear loss, and sensitive symptoms) is lacking. Considering the "educational" aim of this report, the authors should add a paragraph regarding these points.
Author Response
REVIEWER 2
We would like to thank you and the reviewer 1 for the thoughtful feedback provided on our Manuscript entitled “Intracranial epidermoid cyst mimics musculoskeletal shoulder disease: findings from a case report in physiotherapy clinical practice”. Thank you for the opportunity to revise and re-submit.
We highly appreciated the valuable feedback and suggestions of the reviewer. We made every effort to respond to his/her comments and suggestions. Below, we have provided a point-by-point response to each of the issues raised. Within the manuscript itself, all changes to the paper, tables, or images have been highlighted in yellow.
This is an interesting case report with prevalent relevance for clinical practice of physical therapists, describing a case of intracranial epidermoid cyst causing shoulder pain. An accurate physical examination by the PT was able to detect red flags, causing referral to MRI and consequent surgery.
Although the take-home message is clear and nicely discussed, there are some points that need further clarification.
REPLY: we thank Reviewer 2 for appreciating our work.
- Table 1 is misleading in the way it was conceived. The authors should specify that the correlation between site of localization and symptoms is only based on previous reports, as for example a compressive lesion in the right parietal lobe could theoretically cause a wide range of symptoms other than epilepsy.
REPLY: Thank you for this suggestion. We agreed and added a sentence for a better clarification.
- Figure 1: precise timing between consequent events should be specified in the timeline.
REPLY: thank you for this note. We amended accordingly.
- Paragraph 3.6 Referral: a neurological physical examination and precise quantitative EMG/ENG data are lacking from case description. In particular, electromiographic and nerve conduction findings should be described separately and with greater detail.
REPLY: Thank you for this suggestion aimed to improve our manuscript. We agreed with this note.; therefore, we added a new paragraph which better describes the EMG/ENG procedures and obtained data.
- Figures 4 and 5: legends should specify which sequences are reported. Please, consider using the same sequences for pre- and post-operative scans.
REPLY: Thank you for this suggestion, we added the sequences
- Discussion: a neuroanatomical explanation of why intracranial cyst of the posterior fossa should cause shoulder pain (alongside with motor dysfunction, muscle atrophy, hear loss, and sensitive symptoms) is lacking. Considering the "educational" aim of this report, the authors should add a paragraph regarding these points.
REPLY: Thank you for this suggestion. We added a paragraph into the manuscript, accordingly
Reviewer 3 Report
Known in the field based on previous literatures:
1. Specific Epidermoid brain cysts or intracranial epidermoid cyst is congenital in origin and kind of benign intracranial tumors usually form in the very early stages of the development of embryo or the cysts may develop later in life when an injury causes epithelial cells to be trapped in brain tissue.
2. The symptoms of intracranial epidermoid cysts may include hearing loss, severe and involuntary face pain, ringing in the ears, and headaches
3. Epidermoid brain cysts may be detected by MRI and CT scans and treatment usually involves surgery.
In this case study authors reported following findings:
I have gone through the case study titled "Intracranial epidermoid cyst mimics musculoskeletal shoulder disease findings from a case report in physiotherapy clinical practice'. The case report describes the physical examination and clinical decision-making procedures in a patient with an intracranial epidermoid cyst mimicking a musculoskeletal disorder of the shoulder. To investigate the musculoskeletal pain (pain and disability in left shoulder) in a 42-year-old man, the physiotherapist (PT) performed the physical examination and led to hypothesize that the signs and symptoms of pathologies are outside the PT’s scope of practice and refer to another health professional for further investigation.
On the basis neuro-musculoskeletal and cranial nerve assessment, PT referred the patient to a neurosurgeon who prescribed an MRI and an electromyography and detected epidermoid cyst. Neurosurgeon performed craniotomy of the left temporal bone and excised the exposed tumor and further rehabilitation assessment was executed by PT.
The data presented are interesting and generally supportive of the conclusions drawn. The following minor suggestions if incorporated could help in the better understanding of the significance of the work and implications.
Minor/ major Concerns:
1. Explain, how your study is different from rest and how does it address a specific gap in the field?
2. How much similarity and dissimilarity in sign and symptoms you have observed in the current case study as compared to previous reported cases? Please discuss it in discussion.
Author Response
REVIEWER 3
We would like to thank you and the reviewer 1 for the thoughtful feedback provided on our Manuscript entitled “Intracranial epidermoid cyst mimics musculoskeletal shoulder disease: findings from a case report in physiotherapy clinical practice”. Thank you for the opportunity to revise and re-submit.
We highly appreciated the valuable feedback and suggestions of the reviewer. We made every effort to respond to his/her comments and suggestions. Below, we have provided a point-by-point response to each of the issues raised. Within the manuscript itself, all changes to the paper, tables, or images have been highlighted in yellow.
Known in the field based on previous literatures:
- Specific Epidermoid brain cysts or intracranial epidermoid cyst is congenital in origin and kind of benign intracranial tumors usually form in the very early stages of the development of embryo or the cysts may develop later in life when an injury causes epithelial cells to be trapped in brain tissue.
- The symptoms of intracranial epidermoid cysts may include hearing loss, severe and involuntary face pain, ringing in the ears, and headaches
- Epidermoid brain cysts may be detected by MRI and CT scans and treatment usually involves surgery.
In this case study authors reported following findings:
I have gone through the case study titled "Intracranial epidermoid cyst mimics musculoskeletal shoulder disease findings from a case report in physiotherapy clinical practice'. The case report describes the physical examination and clinical decision-making procedures in a patient with an intracranial epidermoid cyst mimicking a musculoskeletal disorder of the shoulder. To investigate the musculoskeletal pain (pain and disability in left shoulder) in a 42-year-old man, the physiotherapist (PT) performed the physical examination and led to hypothesize that the signs and symptoms of pathologies are outside the PT’s scope of practice and refer to another health professional for further investigation.
On the basis neuro-musculoskeletal and cranial nerve assessment, PT referred the patient to a neurosurgeon who prescribed an MRI and an electromyography and detected epidermoid cyst. Neurosurgeon performed craniotomy of the left temporal bone and excised the exposed tumor and further rehabilitation assessment was executed by PT.
The data presented are interesting and generally supportive of the conclusions drawn. The following minor suggestions if incorporated could help in the better understanding of the significance of the work and implications.
REPLY: We thank to Reviewer 3 for the appreciation to our manuscript.
Minor/major Concerns:
- Explain, how your study is different from rest and how does it address a specific gap in the field?
REPLY: The patient reported suffering from chronic neck pain; however, in the last month the neck pain was reported to have changed, becoming persistent and associated with intermittent paresthesia on the upper trapezius and, on the back, on the scapular area, warned as a pinprick sensation both during everyday activities (e.g., lifting c. 5 kilograms objects), and while resting in bed.
Moreover, the pain gradually increased (5/10 NPRS) and diffused wrapping upper trapezius and the entire shoulder complex. The pain increased overtime and progressively the arm function decreased: originally, the patient felt that the arm had only weakened but later he was no longer able to raise the arm over his head.
The initial clinical presentation suggested frequent and fairly common features of neck and shoulder pain. In order to exclude potential or significant or potential “red flags”, the use of a widely and accepted system to rule out serious pathologies or systemic symptoms (fever, weight loss, balance problems) through checklist was performed. Only with this further investigation procedures, the PT was able to recognize for mild intermittent paresthesia of the left cheek and jaw and that he noticed a progressive reduction in hearing in his left ear.
Thus, clinical presentation of our patient seems to be extremely rare and atypical when compared to other significant literature evidence, in which neuromusculoskeletal symptoms are hardly never reported (see Table 1 within the manuscript). This unusual presentation, with the predominance of upper limb dysfunction respect to other more common central alteration (eg: headache, dizziness, nausea) is certainly the main different aspect of our case report. Moreover, being to solve this specific case, better and more refined screening abilities were necessary, highlighting the importance of PT’s expertise in cranial nerve screening; these skills are actually underestimated in clinical practice.
- How much similarity and dissimilarity in sign and symptoms you have observed in the current case study as compared to previous reported cases? Please discuss it in discussion.
REPLY: Thank you for this further consideration. Authors think that the combination of subtle facial reported symptoms and important loss of function of the upper limb mimicking much more common musculoskeletal disorders, are dissimilar from what is reported in literature. The most plausible mechanism which could justify this atypical presentation is described below and was added in the manuscript.
The cyst significantly compressed the brain stem at the posteromedial level. The compression of this area caused the alteration of part of cranial nerve function, specifically the vestibular-cochlear nerve with consequent onset of hearing loss, the facial nerve causing the appearance of check and jaw dysesthesia, and the spinal accessory nerve. This latter, which innervates the trapezius muscle, working in synergy with the serratus anterior muscle, when involved by compression, caused an alteration of shoulder biomechanics, triggering an imbalance between the trapezius and the serratus anterior muscles. This led to a compression of the structures, present at the level of the coraco-acromial arch, which probably caused the onset of shoulder pain. This particular combination of symptoms, at the best of knowledge of the authors, was not yet reported.
Round 2
Reviewer 1 Report
Thanks for your reply.
I agree that epidermal cyst events are rare. However, shoulder pain in patients with epidermoid cysts is not considered a rare condition.
Author Response
REVIEWER 1
I agree that epidermal cyst events are rare. However, shoulder pain in patients with epidermoid cysts is not considered a rare condition.
RESPONSE: We thank Reviewer 1 for this comment. Authors know that shoulder pain in patients with epidermoid cyst was yet reported in literature (DOI: 10.1111/j.1445-2197.1986.tb06165.x; doi: 10.1159/000315559); however, Authors think that the combination of subtle facial reported symptoms and important loss of function of the upper limb are dissimilar from what is usually reported in literature. Infact, is this particular combination of symptoms (for first time noticed in physical therapy direct acces setting), that, at the best of knowledge of the Authors, was not yet reported. Notably, all other causes of epidermoid cyst found in the brain, mainly reported neurological symptoms (please see Table 1 of the manuscript).
Reviewer 2 Report
The authors have addressed my previous concerns in a satisfactory way. The main concern I still have regards the legends of MRI figures, as I think that the left image in Figure 4 might actually be a post-contrast T1-weighted sequence (and not a PD sequence, as reported). Please, check also other figures. Also, be aware of the spelling errors (e.g., disfunction instead of dysfunction).
Author Response
REVIEWER 2
The authors have addressed my previous concerns in a satisfactory way. The main concern I still have regards the legends of MRI figures, as I think that the left image in Figure 4 might actually be a post-contrast T1-weighted sequence (and not a PD sequence, as reported). Please, check also other figures. Also, be aware of the spelling errors (e.g., disfunction instead of dysfunction).
RESPONSE: Thank you for this note. Authors agree with the reviewer 2 and amended the captions. Moreover, we checked for spelling errors.